# Impact of Local High Doses of Radiation by Neutron Activated Mn Dioxide Powder in Rat Lungs: Protracted Pathologic Damage Initiated by Internal Exposure

**DOI:** 10.3390/biomedicines8060171

**Published:** 2020-06-23

**Authors:** Kazuko Shichijo, Toshihiro Takatsuji, Zhaslan Abishev, Darkhan Uzbekov, Nailya Chaizhunusova, Dariya Shabdarbaeva, Daisuke Niino, Minako Kurisu, Yoshio Takahashi, Valeriy Stepanenko, Almas Azhimkhanov, Masaharu Hoshi

**Affiliations:** 1Department of Tumor and Diagnostic Pathology, Atomic Bomb Disease Institute, Nagasaki University, 1-12-4 Sakamoto, Nagasaki 852-8523, Japan; niino-daisuke@nagasaki-u.ac.jp; 2Faculty of Environmental Studies, Nagasaki University, Bunkyo 1-14, Nagasaki 852-8521, Japan; takatsuj@nagasaki-u.ac.jp; 3Dерartmеnt оf Рathоlоgical Anatоmy and Fоrеnsic Mеdicinе, Semey State Medical University, Republic of Kazakhstan, Abay Str., 103, Semey 071400, Kazakhstan; zhaslan_love@mail.ru (Z.A.); craniosex@mail.ru (D.U.); n.nailya@mail.ru (N.C.); dariya_kz67@mail.ru (D.S.); 4Department of Earth and Planetary Science, Graduate School of Science, The University of Tokyo, 7-3-1 Hongo, Bunkyo-ku, Tokyo 113-0033, Japan; minako-kurisu@eps.s.u-tokyo.ac.jp (M.K.); ytakaha@eps.s.u-tokyo.ac.jp (Y.T.); 5A.Tsyb Medical Radiological Research Center—National Medical Research Center of Radiology, Ministry of Health of Russian Federation, 249036 Obninsk, Russia; valerifs@yahoo.com; 6National Nuclear Center of the Republic of Kazakhstan, Kazakhstan 071100, Kazakhstan; ALM_AZ@mail.ru; 7The Center for Peace, Hiroshima University, Higashi-senda-machi, Naka-ku, Hiroshima 730-0053, Japan; mhoshi@hiroshima-u.ac.jp

**Keywords:** internal radiation exposure, ^56^Mn, micro-dosimetry, lung, A-bombing, elemental imaging, apoptosis, elastin, SR-XRF, X-ray absorption near-edge structure (XANES)

## Abstract

Internal radiation exposure from neutron-induced radioisotopes environmentally activated following atomic bombing or nuclear accidents should be considered for a complete picture of pathologic effects on survivors. Inhaled hot particles expose neighboring tissues to locally ultra-high doses of β-rays and can cause pathologic damage. ^55^MnO_2_ powder was activated by a nuclear reactor to make ^56^MnO_2_ which emits β-rays. Internal exposures were compared with external γ-rays. Male Wistar rats were administered activated powder by inhalation. Lung samples were observed by histological staining at six hours, three days, 14 days, two months, six months and eight months after the exposure. Synchrotron radiation—X-ray fluorescence—X-ray absorption near-edge structure (SR–XRF–XANES) was utilized for the chemical analysis of the activated ^56^Mn embedded in lung tissues. ^56^Mn beta energy spectrum around the particles was calculated to assess the local dose rate and accumulated dose. Hot particles located in the bronchiole and in damaged alveolar tissue were identified as accumulations of Mn and iron. Histological changes showed evidence of emphysema, hemorrhage and severe inflammation from six hours through eight months. Apoptosis was observed in the bronchiole epithelium. Our study shows early event damage from the locally ultra-high internal dose leads to pathogenesis. The trigger of emphysema and hemorrhage was likely early event damage to blood vessels integral to alveolar walls.

## 1. Introduction

Radiation from neutron-induced radioisotopes in soil, dust and other materials should be considered to fully understand the radiation effects on the survivors of the Hiroshima and Nagasaki atomic bombings as well as those affected by nuclear power plant accidents and other nuclear disasters involving the scattering of hot particles [1], in addition to the initial radiation directly received from the bombs or other sources. This may be more important for evaluating the radiation risks to the people who moved to these cities soon after the detonations or other nuclear disasters and probably inhaled activated radioactive “dust”. ^56^Mn is known to be one of the dominant radioisotopes produced in soil by neutrons (DS86 Vol. 1 Page 233 [2]). Due to its short physical half-life of 2.58 h, ^56^Mn emits radiation during substantially only the first h after inhalation.

We investigated the biologic effects of internal exposure by ^56^MnO_2_ powder on rats compared with externally exposed to ^60^Co γ-ray and control groups [3,4]. The long duration and severe effects of the internal exposure group with even the short half-life of ^56^Mn and comparatively mild effects of the external whole-body exposure group led us to a need to determine initial effects of the early exposure event.

A clear picture of the early stage events should be critically important in establishing the difference of the effects of internal and external exposure. The impact of internal exposure to organ tissue at low tissue average doses was equivalent to high dose effects from external exposure. The previous findings were shown in the data of lung tissue for hemorrhage, emphysema and inflammation of three days and two months after 100 mGy internal exposure [3,4]. Here we present early stage data of six hours after exposure of ^56^MnO_2_ powder and late stage data of six months after the exposure, including additional data of the higher internal exposure. Understanding the possibility of potential late effects occurring after a succession of cell turnover with the early stage internal β-ray exposure by ^56^Mn with its short physical half-life, leads to a more complete picture of the pathologic effects of internal radiation.

Moreover, we present X-ray spectroscopic imaging data of a tissue sample taken in the area of maximal concentration of Mn. The method employed here was synchrotron radiation X-ray fluorescence mapping using X-ray microbeam (SR-XRF) with speciation of Mn at the point of interest by X-ray absorption near-edge structure (XANES) spectroscopy. The results showed that ^56^MnO_2_ particles being embedded and ejected from lung tissue and the affected tissue around the embedded particles.

The scope of the project was limited by the difficulty of activation of Mn powder and logistical management of biologic samples. However, we feel the importance of the data, showing the significant pathologic impact of radioactive hot particles, justifies its presentation in spite of the small sample sizes.

## 2. Materials and Methods

### 2.1. Chemicals and Radiations

The size distribution of MnO_2_ powder particles used in this experiment is same as that previously reported on [5]. MnO_2_ powder containing ^56^Mn (T1/2 = 2.58 h) was produced by neutron activation of 100 mg of ^55^MnO_2_ powder (Rare Metallic Co., Ltd., Japan) at the IVG.1 M (“Baikal-1”) nuclear reactor [6] using a neutron fluence of 4 × 10^14^
*n*/cm^2^ and an irradiation time of 2000s (1×), 4000 s (2×), 8000 s (4×) (Table 1) [5]. Briefly, activated powder with ^56^Mn activities of 2.74 × 10^8^ Bq (^56^Mn(1×)), 2 × 2.74 × 10^8^ Bq (^56^Mn(2×)) and 4 × 2.74 × 10^8^ Bq (^56^Mn(4×)) was sprayed pneumatically over rats located in an experimental box. γ 2.0 Gy group was externally exposed to 2.0 Gy of ^60^Co γ-ray at a dose rate of 2.6 Gy/min using a Teragam K2 unit (UJP Praha, Praha-Zbraslav, Czech Republic). Control group was unexposed. The initial specific activities (activity per mass) of neutron-activated MnO_2_ powder were 4× higher in ^56^Mn(4×) group and 2x higher in ^56^Mn(2×) group. The average radiation doses were estimated in the same way as Stepanenko et al [5]. The average radiation doses of each organ received in ^56^Mn(4×) and ^56^Mn(2×) groups were substantially almost same or lower than those received in ^56^Mn(1×) group, i.e., the lung received 0.10 Gy in ^56^Mn(1×) while 0.11 Gy in ^56^Mn(4×) group and 0.051 Gy in ^56^Mn(2×) group.

### 2.2. Animals and Treatment

Ten-week-old male Wistar rats were purchased from Kazakh Scientific Center of Quarantine and Zoonotic Diseases, Almaty, Kazakhstan. They were housed in plastic cages under climate-controlled conditions at 22 ± 2 °C with a relative humidity of 50% ± 10% and a constant day/night cycle (light 0.700–19.00 h). They were maintained with free access to basal diet and tap water. For the 8-month study, ^56^Mn(1×) rats were compared with a group of rats exposed to Mn-stable (not activated), a group to ^60^Co; externally-exposed-to-2.0-Gy ^60^Co-γ-ray and an unexposed control group. All applicable international, national, and/or institutional guidelines for the care and use of animals were followed. The animal experiment was approved by the Animal Experiment committee of Semey Medical University, Republic of Kazakhstan and conducted in accordance with the Institutional Guide for Animal Care and Use (18 April 2014).

Pathologic lung samples of ^56^Mn(2×) group and ^56^Mn(4×) group were observed at 6 h, 3 days, 14 days, 2 months 6 months and 8 months after exposure. There were 3–6 rats in each group.

### 2.3. Pathology

Whole lungs were collected, dissected and fixed in 10% neutral buffered formaldehyde and embedded in paraffin. Sections of 4 µm thickness were prepared and stained with H&E. For pathologic examination of the lung tissue, the grades of hemorrhage, emphysema and inflammation (inflammatory cell population) were scored from “−” to “+++” (Table 2) [3,4]. Pathologic features were graded by three independent observers semiquantitatively according to the four-tiered scoring system, 0: negative (−); 1: weakly positive (+); 2: moderately positive (++); and 3: strongly positive (+++), for each of three histological features hemorrhage, emphysema and inflammation. Additionally, atelectasis, pneumonia and granuloma were graded. Elastin staining was done using EVG and collagen staining was achieved using AM. Percentage of positive areas were analyzed by Olympus cellSens Dimension using 5 images per sample. Apoptotic cells were stained by TUNEL method (ApopTag Peroxidase In Situ Apoptosis Detection Kit, S7100, Chemicon) as described, by the manufactures’ suggested protocol [7].

### 2.4. SR-XRF-XANES Analysis

XRF analyses combined with XANES spectroscopy were conducted at BL-4A of the Photon Factory in the High Energy Accelerator Research Organization (KEK-PF, Tsukuba, Japan). All the experiments were carried out in the top-up running mode of PF at room temperature in the ambient air, and the basic processes of the analysis are similar to those described in Takahashi et al. [8]. The energy of the incident X-ray was 12.9 keV. The X-ray was focused into 5 (horizontal) × 5 (vertical) μm^2^ at BL-4A using Kirkpatrick-Baez mirror optics. A thin-section sample of the specimens was fixed on a sample holder oriented at 45° to the X-ray beam. The sample was irradiated by the micro-focused X-ray and the specimen stage was scanned in the X–Y directions, two-dimensionally, to obtain the areal elemental distribution images using intensities of XRF from each element detected by a silicon drift detector (SDD). The size of the scanned areas varied within several millimeters and the scanning steps varied from 5 μm to 50 μm. The obtained XRF spectra at BL-4A were processed using PyMca software (Version 4.7.3). In addition, the XRF spectra were measured for 300 s at the spots containing metallic elements [9]. Chemical species of Mn such as valence of Mn at the point of interest (POI) found by SR-XRF analysis was determined by Mn K-edge XANES spectroscopy. The uncertainty in energy is 0.1 eV. The spectra at the POI were compared with those of reference compounds including MnO_2_, Mn_2_O_3_, Mn_3_O_4_ and MnCO_3_ to estimate average valence of Mn at the POI. The XANES spectra of Mn were processed using a XAFS data analysis software, REX2000 (Rigaku Co. Tokyo, Japan).

### 2.5. Mn particle Dose Rate

The electron energy spectrum of ^56^Mn β-ray in tissue surrounding the ^56^Mn particles was calculated and the β-ray absorbed dose rate was calculated from ^56^Mn β-ray energy spectrum [10] and electron stopping powers [11] according to the activity levels immediately following intake of the ^56^Mn particles. The density of MnO_2_ particles and the tissue were assumed 5.026 g/cm^3^ and 1 g/cm^3,^ respectively and the atomic composition of them was assumed for both equivalent to water for the calculation. The effect of the different assumption from the real composition for the particles is negligible compared to the calculated results because the size of the particles is small relative to the range of the β-ray. Accumulated dose was also calculated to 6 h and the time that the activity was almost completely attenuated. The half-life is only 2.58 h and the activity was almost totally attenuated at the time of fixation except 6 h (Table 3).

### 2.6. Statistical Analysis

All values were expressed as the mean ± standard error (SE) involving three to six animals. Mann–Whitney U test was applied to evaluate the statistical significance of difference between groups.

## 3. Results

### 3.1. Early Event Damage to Lung Tissue for Internal Exposure Is Uncharacteristically Rapid and Severe

The histological changes in the lung for the ^56^Mn(1×) (1 × 2.74 × 10^8^ Bq) exposed rats, showed evidence of emphysema, hemorrhage and severe inflammation from 6 h through 6 months, which was more pronounced in ^56^Mn(4×) (4 × 2.74 × 10^8^ Bq) group. In γ 2.0 Gy group only minimal inflammation was observed and control group showed no changes. Observation at 6 months post exposure revealed rats in groups ^56^Mn(4×) and ^56^Mn(2×) (2 × 2.74 × 10^8^ Bq) with damage in lung tissue having pathologic grades similar or greater than those observed at day 3. Prominently differing from γ 2.0 Gy group and control.

The internally exposed animals had higher levels of hemorrhage, emphysema and inflammation compared to the externally exposed animals and control. Hemorrhage for both 2× and 4×^56^Mn exposed rats peaked at 3 days after exposure whereas the externally exposed group returns to control levels at 3 days. Emphysema grew progressively worse in a dose dependent manner through 14 days post exposure for the internally exposed groups, while the externally exposed group showed a steady decline with the passage of time. In the internally exposed groups, inflammation was observed from 6 h post exposure and continued over the long term, while the externally exposed group normalized to control levels (Table 2, Figure 1, Figure 2 and Figure 3).

At 6 h, Verhoeff–Van Gieson (EVG) stained elastic fibers show abnormally localized expression surrounding the alveoli in a staggered distribution, broken at intervals in ^56^Mn treated samples (Figure 4A). γ 2.0 Gy shows a similar effect at a lower expression level of elastin, compared with 4×^56^Mn (Figure 2D). γ 2.0 Gy samples normalize to control levels after 3 days, whereas ^56^Mn remains high through 6 months. Lower expression levels of Azan–Mallory (AM) stained collagen can be seen in ^56^Mn(4×) group at 6 h, rising at 3 days and remaining at control level (Figure 2E). ^56^Mn(2×) group showed only low collagen at 2 month, whereas elastin expression was high at 2 month, continuing to 6 months. Apoptosis is clearly evident in the bronchiole epithelium in ^56^Mn group exclusively at 3 days, 14 days, 2 months and 8 months. The prominent data were for 2 months. No occurrence was observed in other groups (Figure 4A,D).

### 3.2. Late Histology with Prominent Atelectasis and Pneumonitis for both 2× and 4×^56^Mn Exposure, Granuloma and Severe Hemorrhage

Late effect pathologic findings showed atelectasis, granuloma and severe pneumonitis at 6 months, resulting in continued tissue damage from emphysema and long-term inflammation. No thickening of alveolar walls and no gradual development of emphysema was observed with the internal exposure (Figure 3).

As can be seen in Figure 1, early effects presage late effects. High score levels for hemorrhage, emphysema and inflammation at both 2× and 4× radiation at day 3 continued to month 2 post-irradiation. Figure 3C,D showing late histology with prominent atelectasis and pneumonitis for both 2× and 4× exposure, granuloma (Figure 3E) and severe hemorrhage (Figure 3F), at 4× exposure. Figure 3A,B, control and γ 2.0 Gy, respectively, showed no pathologic findings.

Table 2 and Figure 4 show pathologic findings at 8 months. There was no hemorrhage or inflammation for control group. Emphysema is significant for ^56^Mn group. ^56^Mn shows substantially higher inflammation compared with Mn stable and control. EVG staining shows 4 times higher expression of elastin fiber for ^56^Mn compared to control, γ 2.0 Gy and Mn stable (Figure 4A,B). No significant difference can be seen between Mn stable, ^56^Mn and control for collagen expression, whereas 2.0 Gy γ collagen expression is significantly higher than control (Figure 4A,C).

### 3.3. SR-XRF Analysis Revealed that Masses of Manganese, Iron and Calcium Attached to Damaged Tissue

Shot1 and Shot2 were identified as condensed accumulations of manganese and iron (Figure 5A,B,D,E). From the elemental distribution, we estimated the elementary profile for Shot1 and Shot2. Maximum Mn and Fe concentrations of the scan site Shot1 and Shot2 was roughly estimated using average Ca peak count of the normal tissue part of Shot1 and already known Ca standard concentration in the normal tissue for calibration.

The focus of Shot1 is the brown square-shaped particle, a condensed accumulation of manganese and iron. It is located at the center of a respiratory bronchiole. The somewhat anomalous presence of the particle may have originated from a larger mass of manganese and iron in the process of physical excretion via phlegm or other matter. The focus of Shot2 is a significantly larger particle detached from the tissue of the alveolus and stuck adjacent the original position. It seems to be a mass of manganese, iron and calcium displaced with damaged tissue.

### 3.4. Dose Rate and Accumulated Dose Are Functions of Diameter and Distance

Dose rate and accumulated dose by electron (β-ray) emitted from MnO_2_ particle are functions of diameter and distance from the center. The dose rate depended on the diameter of the particle. Larger diameter particles gave higher dose rates. Table 3 shows the absorbed dose rate in tissue encircling the activated MnO_2_ particles. Dose rate was calculated according to activity levels immediately following intake of neutron-activated MnO_2_ particles. A 5 µm diameter particle has a 2.70 Gy/h absorbed dose rate and 8.05 Gy accumulated dose for 6 h at 2.5 µms from the center of the tissue encircling the particle. The accumulated dose of tissue is shown in Figure 5B and is 2× of that shown in Table 3, the accumulated dose at 2.5 μm distance radially from the center of the MnO_2_ particle is calculated as 16.1 Gy, at 5 μm—2.82 Gy, 10 μm—0.662 Gy and 50 μm—0.024 Gy. Shot1 located in the bronchiole. Shot2, located in the damaged tissue of the alveolus was identified within 0.024 Gy circle at 50 μm distance from the center. The sample was taken 6 h after exposure to observe the early event of internal exposure. We have already reported the size distribution of MnO_2_ powder particle used in this experiment [5].

### 3.5. XANES Spectroscopy for the Analysis of Particles Embedded in Lung Tissue Samples (Shot1 and Shot2), They Were Determined to Be Mn

Utilizing XANES spectroscopy for the analysis of particles embedded in lung tissue samples (Shot1 and Shot2 in Figure 5), they were determined to be Mn. XANES spectroscopy exhibits relative energy position and intensity of Mn compounds, and therefore we were able to distinguish these samples as metabolized Mn [12]. A ^56^Mn particle that has a valence of 4 chemical species as MnO_2_ changed to a valence of 2 after deposition in lung tissue (Figure 6).

## 4. Discussion

The delayed effect of acute external radiation exposure was noted in a study by MacVittie et al. [13]. 10.74 Gy whole thorax and lung external exposure of high energy photons resulted in similar late pathologic findings to the ^56^Mn internal exposure of our study, pneumonitis and inflammation. According to MacVittie et al. it is with predominant neutrophils, macrophages and a thickening of alveolar wall. The internal exposure, moreover, prevalently resulted in an increase of pulmonary emphysema and atelectasis with hemorrhage at month 6. Our results indicate an increasing possibility of respiratory failure, morbidity and mortality from the internal exposure.

Early event damage to lung tissue for the internal exposure is uncharacteristically rapid and severe compared with external radiation exposure. Emphysema was not observed in the ^60^Co γ exposed group in our study except at six hours, nor even at the significantly higher dose of 10 Gy in Macvittie et al. [13]. No thickening of alveolar walls and no gradual development of emphysema was observed with the internal exposure, suggesting a unique internal radiation pathology. Pulmonary emphysema is characterized histologically by the destruction of alveolar walls and enlargement of air spaces by lung epithelial cell apoptosis [14], and in cigarette smoking by oxidative stress, cell death by apoptosis, and senescence with decreased repair [15,16]. This is a similar mechanism to models of apoptosis-dependent emphysema [17,18] or emphysema following high-dose ionizing radiation exposures [19,20]. Our findings suggest stem cell apoptotic damage for internal exposure. Apoptosis was clearly evident in the bronchiole epithelium in the ^56^Mn group at two months. Comparably, Homma et al. have reported renal toxicity to uranium exposure in relation to uranium distribution and uranium-induced apoptosis in the kidneys [21,22,23].

In contrast with internal exposure, for external exposure, there are many papers that provide background on the mechanisms underlying the pathogenesis of radiation pneumonitis and radiation pulmonary fibrosis. Early external radiation-induced lung damage leads to direct-endothelial damage and vasodilatation, as well as changes in epithelial lung components, particularly the presence of Type II pneumocytes [24,25]. Recently, in Groves et al. [26], predictive data were shown to evaluate pulmonary epithelial damage in several radiation lung injury models. There were elevations of the circulating ratio of club cell secretory protein to surfactant protein D levels, expressed by epithelial club cells and Type II pneumocytes in stem cell dysfunction. Lung injury models revealed differential regenerative capacities of epithelial cells of the bronchioles [27]. Stem cell message reduced in the lung with both internal and 5 Gy external exposure [28]. Basal cells in trachea and bronchi may be stem cells or progenitor cells for the trachea-bronchiolar epithelium with a cell turnover time of about 20–30 days [29,30,31]. Type II pneumocytes and basal cells are candidates for lung stem cells.

Diagnosis of pulmonary pneumonitis related fibrosis is based primarily on collagen deposition. The diagnosis occurs in high dose (5 Gy whole body followed by 10 Gy to the lungs) for externally exposed experiments [26]. Here we showed late stage data of abnormal elastin for internal exposure. Irradiated ^56^Mn groups showed heightened abnormal elastin expression (elastin-signature). Houghton et al. [32] showed that elastin fragments drive disease progression in a murine model of emphysema. Our data suggests the elastin-signature surrounding the alveoli in a staggered distribution, broken at intervals in ^56^Mn treated samples as early as six hours is an additional predictive indicator of radiation induced elastin fibrosis, which occurred in a local dose dependent manner. Therefore, early event data of elastin-signature such as we show in Figure 2D and Figure 4A,B may be predictive of late pneumonitis for internal exposure.

2.0 Gy γ shows a lower level similar effect of elastin expression to ^56^Mn(4×) only at six hours (Figure 2D and Figure 4A,B). The ^56^Mn internal exposure of tissue average 0.11 Gy to the lungs (Table 1) is roughly equivalent effect to an external ^60^Co 2.0 Gy γ at six hours and more severe at later time (Figure 2D). The evidence indicates the cause of the severer effect of the internal exposure is likely the difference of local dose (Table 3). The doses of γ radiation used in our studies were significantly lower (less than 10-fold) than those of radiotherapy, but 10-fold higher than lifetime exposures to γ radiation on Earth. The effect of ^56^Mn internal exposure is similar in effect to that of a 20 times higher dose of γ external exposure. It indicates a high-LET-like property of the internal exposure. The incidence of lung tumors in high LET ^239^PuO_2_ groups was shown to be 21 times higher than that of the groups exposed to the lower LET radiations by Hahn et al. [33]. The ionizations of ^56^Mn internal exposure are mainly due to β-ray and distribute concentrating near the ^56^MnO_2_ particle like the ionizations near an α-ray track. However, the elastin-signature did not occur in externally exposed samples. Along with high levels of emphysema and hemorrhage, the elastin-signatures reveal a unique pathology for internal exposure.

Focal pulmonary emphysema and atelectasis with and without hemorrhage were observed in some of the early A-bomb casualties in Hiroshima by Liebow et al. [34], but it was difficult to interpret the cause because the existence of a substantial effect from internal exposure was unknown at that time. Their data shows effects similar to our data (Figure 20. Group I of Liebow et al., [34], Male, 13 years of age. Bombed at approximately 1200 m from Hypocenter. Died on the third day) (Figure 1A,B). The cause of hemorrhage in the victims has not been well understood. Primarily, hemorrhage has been thought to be due to external exposure. Our findings show lung hemorrhage occurring with a relatively low tissue average dose internal exposure. However, the dose is not homogeneous, but ultra-high locally. We have reported previously that the absorbed dose of cell nuclei penetrated with an alpha particle is as high as 4 Gy for blood vessels and 2 Gy for lung cells [1]. Lundgren et al. [35] found a significant relationship between the progression of pulmonary emphysema and the deposition, distribution and retention of ^239^PuO_2_ particles. The onset of pulmonary emphysema should be related to local dose rather than tissue average dose.

Particles were identified (Shot1 and Shot2) as Mn^2+^ using XANES spectroscopy. A ^56^Mn particle that had a valence of four chemical species as MnO_2_ changed to a valence of two after deposition in lung tissue. The elementary profile of Shot2 is closer to the standard profile for lung tissue which suggests that it is a detached piece of lung tissue, whereas Shot1, being composed largely of Fe and located in a bronchiole cavity, is likely to be an ejected blood clot around Mn particle. In contrast to the standard elementary profile, Shot2—and in particular Shot1—include a large percentage of Fe likely owing to hemorrhage in the early event. The focus of radiation induced pathology has primarily been on the long-term effects of external exposure. Our findings point to the need for a better understanding of the impact of highly localized early effects of internal exposure and the pathologic chain of events initiated by them.

The compromised structural integrity caused by the early stage elastin dysfunction could play a major role in the breakdown of epithelial repair mechanisms, resulting hemorrhage and the gradual development of emphysema. Loss of elasticity leads to age related phenotypes such as emphysema and arteriosclerosis [36,37]. Rosenbloom et al. [38] have shown elastin’s crucial role in organ structural integrity. ^56^Mn(4×) group resulted severe pathologic findings, atelectasis, pneumonitis and granuloma which were more clearly observed than ^56^Mn(1×) group. The tissue average accumulated absorbed doses are almost same around 0.10 Gy (Table 1). A pathologic progression potentially leading to mortality is not likely dependent on the average accumulated dose level, but rather on specific activity of the ^56^Mn particle. The local accumulated dose of the point neighboring a five-micrometer-diameter radioactive particle was as high as 8.05 Gy (^56^Mn(1×)) and 32.2 Gy (^56^Mn(4×)) at six hours post inhalation. Early event damage from locally ultra-high internal dose leads to pathogenesis.

The pathogenetic progression initiated by early cellular damage from radiation exposure has a greater impact than has been previously reported about laboratory animals [39,40]. Although these previous researches have shown pathologic changes in the lungs from an external dose of more than 8 Gy, internal exposure can induce pathologic changes at tissue average dose of 0.10 Gy in our experiment. The local accumulated dose of the point proximal to the radioactive particle is as high as 8 Gy (Table 3) and is possibly sufficient to induce pathologic changes in the lung tissue. In the case of human whole body γ exposure, the threshold of pneumonitis is estimated to be 6 Gy in ICRP Pub 103 [41] as a deterministic effect of radiation. Internal exposure pathology is clearly related to the local radiation exposure in a dose dependent manner. We are proposing an alternate pathogenesis of radiation induced pneumonitis characterized by abnormal elastin expression, emphysema and hemorrhage at successive stages interfering with epithelial repair mechanisms. Early onset of a senescence-like effect may be reflected by abnormal elastin expression, occurring with the disruption of repair mechanisms by apoptosis in stem cells located near a hot particle with local high dose internal exposure.

In our experiment, early abnormal elastin expression and evidence of apoptosis presage late effects. Apoptosis occurring in the bronchiole is likely an indicator of stem cell damage [26]. It occurs exclusively in the internally exposed group, even at 0.10 Gy average dose. We identified ^56^Mn particles in a bronchiole cavity and alveolar tissue. Apoptotic stem cell damage for internal exposure continued through two months evidence and it suggests that epithelial repair mechanisms are compromised by damaged DNA. The physical half-life of ^56^Mn is 2.58 h, however, pathologic effects possibly expand with successive abnormal cell divisions with no additional radiation exposure.

Damaged cells undergoing interphase death, i.e., apoptosis and necrosis may contribute to the pathologic progression initiated by internal exposure, as the locally ultra-high dose with internal exposure quite likely exerts an impact comparable to the effect of high LET radiation, which also results in a locally ultra-high dose. Tissue cells damaged severely by external ^60^Co γ-ray exposure may turn over with new differentiated cells because damage of the stem cells may not be severe [42,43]. Therefore, the pathologic condition may diminish gradually. Long term trends of radiation effects on data of A bomb survivors show the initial high incidence and steep decreasing of leukemia deaths and yearly increasing incidence of solid cancer deaths [44]. The time dependent pattern of the first phase leukemia death incidence could correlate to the pattern of elastin expression with external exposure and the pattern of the solid cancers could correlate to that with the internal exposure, which is likely owing to severe damage in stem cells. Embedded hot particles are emitting varying intensities of radiation from low to ultra-high. Tissue surrounding hot particles suffers a variety of injuries resulting from simultaneous exposures to varying internal radiation doses.

The impact of hot particles in the internal exposure, i.e., the early effects, subsequent pathogenic profile and the low average, but locally ultra-high dose has, we believe, significant implications for A-bomb survivors and populations exposed to radioactive plumes or airborne radioactive particles.

Another possibility is the enhancement of biologic effects by the collision of electrons (β-ray) with Mn elements. The collision of high energy charged particles with high-Z element enhances the biologic effects [45,46,47,48,49,50,51]. Therefore, the effects of electrons emitted from ^56^Mn are also possibly enhanced with MnO_2_ particles. However, Z of Mn is 25 and Fe is 26. Fe is a common element in mammalian tissue and the interpretation of the enhancement is not clear. Experiments with γ-rays and nonactivated MnO_2_ would clarify the issue.

## Figures and Tables

**Figure 1 biomedicines-08-00171-f001:**
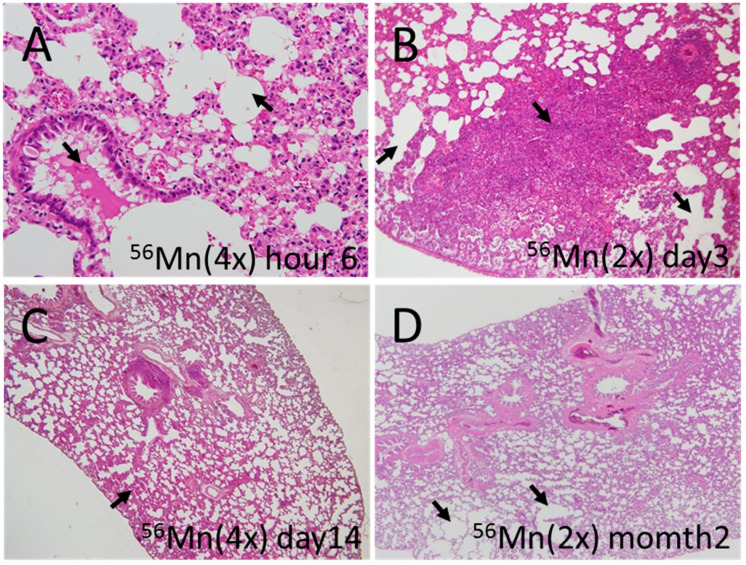
Representative fields of lung tissue early histology collected from ^56^Mn exposed rats. (**A**) ^56^Mn(4×) radiation collected at 6 h post-irradiation. Arrows indicate severe hemorrhage and emphysema; (**B**) ^56^Mn(2×) radiation collected at day 3 post-irradiation. Arrows indicate severe atelectasis and emphysema; (**C**) ^56^Mn(4×) radiation collected at day 14 post-irradiation. Arrow indicate moderate hemorrhage; (**D**) ^56^Mn(2×) radiation collected at month 2 post-irradiation. Arrows indicate severe emphysema. H&E stain, A: 40×, B: 10×, C, D: 4× objective magnification.

**Figure 2 biomedicines-08-00171-f002:**
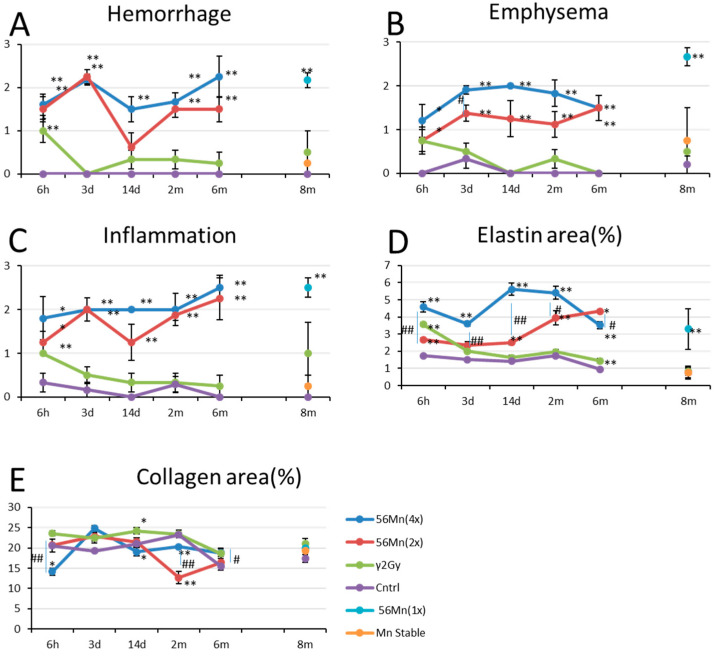
Histological scored findings in the lung in rats exposed to Mn stable, ^56^Mn, γ 2.0 Gy and Control groups. (**A**) shows hemorrhage findings for four groups γ 2.0 Gy, ^56^Mn(2×), ^56^Mn(4×) and control and findings γ 2.0 Gy, Mn stable, ^56^Mn1(×) and control; (**B**) shows findings for emphysema for the same experiments; (**C**) shows the inflammation findings for the same experiments; (**D**) shows the elastin findings for the same experiments; (**E**) shows the collagen findings for the same experiments. Bars, mean ± SEM (*n* = 3–6). * *p*<0.05, ** *p* < 0.01 vs. Control. # *p* < 0.05, ## *p* < 0.01 vs. ^56^Mn(2×).

**Figure 3 biomedicines-08-00171-f003:**
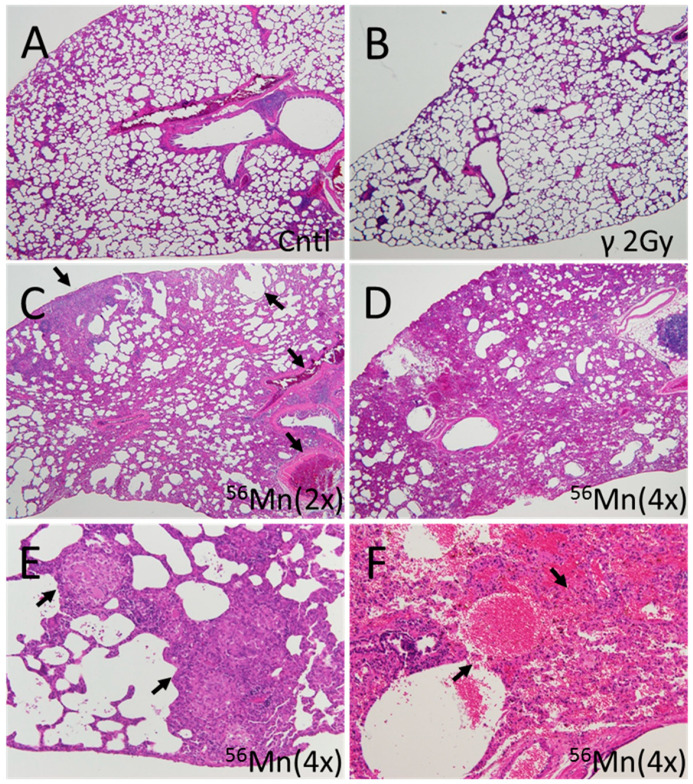
Representative fields of lung tissue late histology from ^56^Mn exposed, 2.0 Gy γ exposed and Control rats collected at month 6 post-irradiation. (**A**) Lung of a control rat. No pathologic changes were observed; (**B**) Lung of a 2.0 Gy γ exposed rat. No pathologic changes were observed; (**C**) Lung of a ^56^Mn(2×) exposed rat. Widespread emphysema and atelectasis as well as hemorrhage with vessel thrombosis were observed (arrows); (**D**) Lung of a ^56^Mn(4×) exposed rat. More extensive damage, pneumonitis resulting from severe inflammation and inflammatory cell infiltration as well as intra-alveolar hemorrhage were observed; (**E**) Lung of a ^56^Mn(4×) exposed rat. Arrows indicate granuloma surrounded by emphysema was observed in a lung lobe; (**F**) Lung of a ^56^Mn(4×) exposed rat. Arrows indicate severe hemorrhage and inflammatory cell infiltration. A higher magnification of D. H&E stain, 4× (A–D) and 20× (E,F) objective magnification.

**Figure 4 biomedicines-08-00171-f004:**
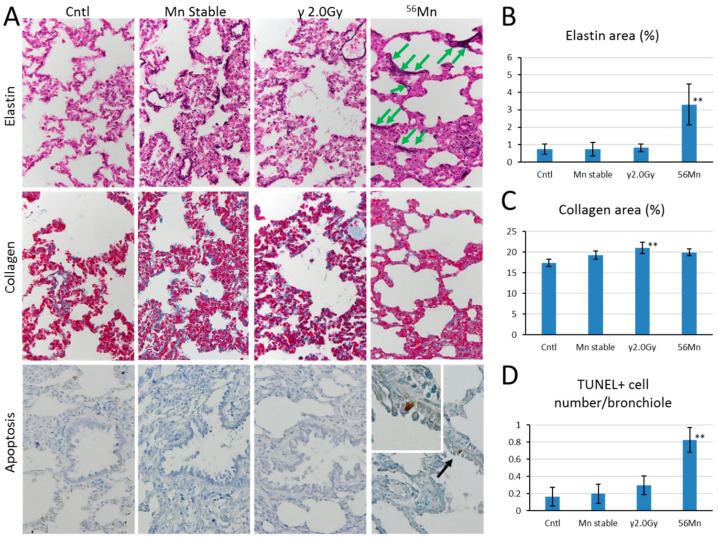
Comparison of elastin and collagen late deposition 8 months and apoptosis effects 2 months after internal (^56^Mn(1×)) and external radiation exposure. (**A**) Lung sections from rats were prepared after exposure and were evaluated for elastin deposition (purple black) identified via Verhoeff–Van Gieson staining and collagen deposition (blue) identified via Azan–Mallory staining and the presence of apoptotic cells (brown, black arrow) in bronchioles identified via TUNEL staining. Broken elastin fibers were observed (green arrows). Original magnification 40×. Representative images of control rat (0 Gy), Mn stable exposure rat, γ 2.0 Gy irradiated rat (2.0 Gy) and ^56^Mn irradiated rat are shown in panels. Radiation induced changes in percentage elastin area (**B**), percentage collagen area (**C**) and number of TUNEL positive cells in bronchioles (**D**), were determined using cellSens Dimension image acquisition and computer-aided quantification of sampled images. Bars, mean ± SEM (*n* = 3–6). Significantly different (** *p* < 0.01) from 0 Gy irradiated controls. 40× (**A**) objective magnification.

**Figure 5 biomedicines-08-00171-f005:**
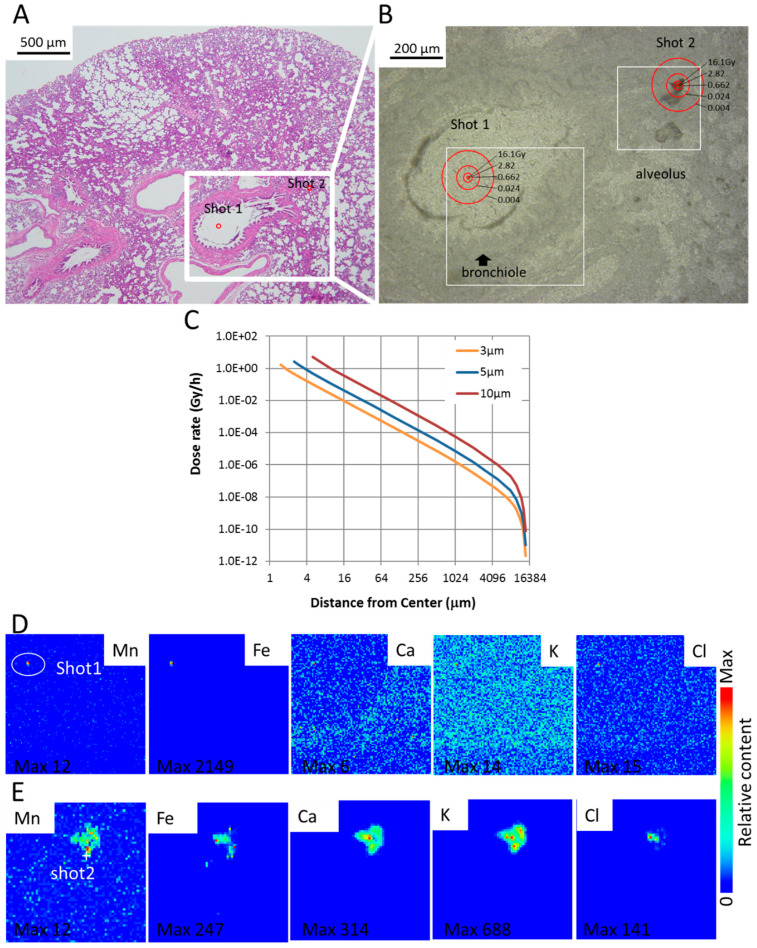
Early elementary profile, histopathologic image and electron dose rate in lung tissue 6 h after ^56^Mn(2×) internal exposure. (**A**) Shot1, located in the bronchiole cavity and Shot2, located in the damaged tissue of the alveolus are observed particles of manganese and iron. H&E stain, 4× objective magnification; (**B**) As can be seen, a 5-μm-diameter particle has a 5.4-Gy/h absorbed dose rate immediately following intake and 16.1-Gy accumulated dose for 6 h at 2.5 μm measured from the center; (**C**) dose rate for ^56^Mn. The dose rate depended on the diameter of the particle (3, 5 and 10 μm). Larger diameter particles showed higher dose rate. Elementary imaging (beam size: 1 μm × 1 μm) of Shot1 (**D**) (5 μm step, 500 × 500 μm) and Shot2 (**E**) (5 μm step, 300 × 300 μm) of boxed areas marked in (**B**) are shown. Maximum concentrations are Shot1; Mn 2900 μg/g, Fe 530,000 μg/g and Shot2; Mn 2900 μg/g, Fe 61,000 μg/g. Corresponding serial section stained with H&E (**A**) and converted to grayscale is shown in (**B**).

**Figure 6 biomedicines-08-00171-f006:**
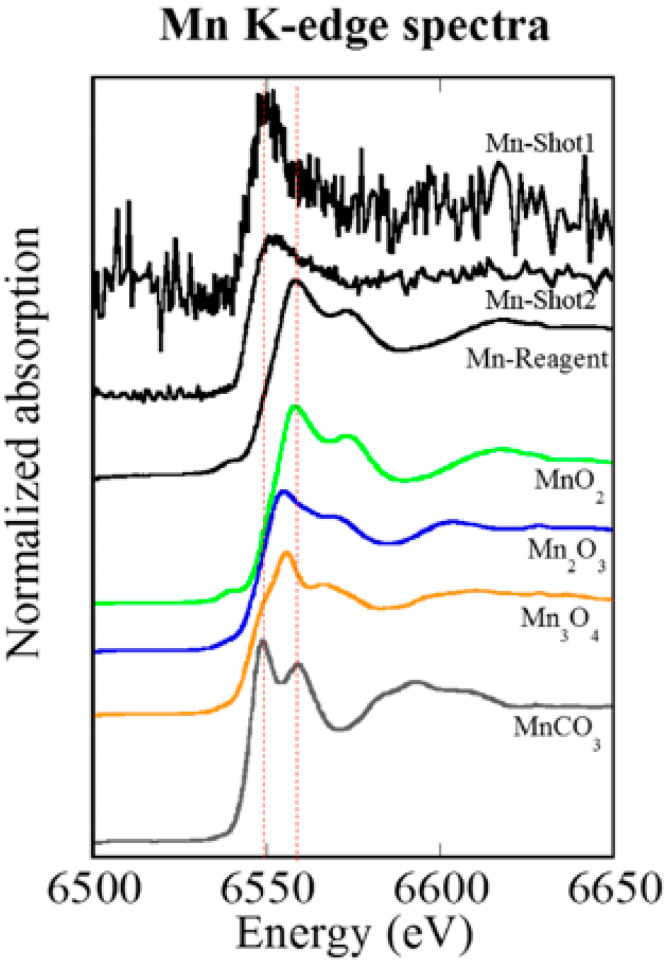
Mn K-edge XANES spectra (in Figure 5), located in the bronchiole cavity (Shot1) and located in the damaged tissue of the alveolus (Shot2), 6 h after ^56^Mn(2×) internal exposure. The Mn K-edge XANES spectra both of Shot1 and Shot2 were different from that of Mn reagent and MnO_2,_ but similar to that of MnCO_3_ solution. The red dotted lines indicate locations of peaks observed in MnCO_3_ solution. The special similarities indicate that Mn is not in the Mn^4+^ (Mn reagent, MnO_2_), but is Mn^2+^(MnCO_3_).

**Table 1 biomedicines-08-00171-t001:** ^56^Mn specific activities *A*_o_ and tissue average accumulated absorbed doses *D* of internal irradiation in different organs and tissues of experimental rats.

Group of Experiment	^56^Mn(1×)						^56^Mn(2×)						^56^Mn(4×)					
Organ	*A*_0_ (kBq/g)			*D* (Gy)			*A*_0_ (kBq/g)			*D* (Gy)			*A*_0_ (kBq/g)			*D* (Gy)		
Liver	4.14	±	0.35	0.015	±	0.001	1.40	±	0.33	0.005	±	0.001	3.00	±	0.70	0.011	±	0.003
Heart	5.46	±	0.60	0.016	±	0.021	2.00	±	0.30	0.006	±	0.001	4.50	±	0.94	0.014	±	0.004
Kidney	3.97	±	0.48	0.013	±	0.002	2.30	±	0.51	0.008	±	0.002	3.60	±	0.60	0.012	±	0.003
Tongue	45.0	±	5.4	0.069	±	0.011	12.0	±	2.8	0.018	±	0.005	28.0	±	5.0	0.043	±	0.012
Lungs	71.8	±	9.3	0.100	±	0.014	37	±	7	0.051	±	0.011	81	±	13	0.110	±	0.023
Esophagus	25.5	±	3.6	0.050	±	0.009	6.4	±	1.6	0.012	±	0.003	16	±	3.3	0.030	±	0.007
Stomach	148	±	16	0.24	±	0.03	140	±	28	0.22	±	0.06	210	±	47	0.33	±	0.08
small intestine	811	±	93	1.33	±	0.17	350	±	79	0.58	±	0.14	890	±	210	1.48	±	0.37
Large intestine	1011	±	100	1.65	±	0.18	410	±	87	0.69	±	0.17	1160	±	270	1.90	±	0.47
Trachea	5.79	±	0.75	0.014	±	0.002	3.90	±	0.77	0.010	±	0.002	6.50	±	1.40	0.016	±	0.004
Eyes	13.2	±	1.7	0.021	±	0.003	19.0	±	4.4	0.031	±	0.008	24.0	±	5.3	0.004	±	0.010
Skin	40.6	±	4.9	0.076	±	0.010	45.0	±	8.2	0.086	±	0.020	49.0	±	9.0	0.095	±	0.020
Whole body	83.4	±	11.0	0.150	±	0.025	51.0	±	11.0	0.091	±	0.026	77.0	±	15.0	0.140	±	0.030
^56^Mn(1×); Initial total activity of 100 mg ^56^MnO_2_ powder sprayed was 2.74 × 10^8^ Bq [5].																		
^56^Mn(2×); Initial total activity of 100 mg ^56^MnO_2_ powder sprayed was 5.48 × 10^8^ Bq.																		
^56^Mn(4×); Initial total activity of 100 mg ^56^MnO_2_ powder sprayed was 1.10 × 10^9^ Bq.																		

**Table 2 biomedicines-08-00171-t002:** Histological findings in the lung in rats exposed to ^56^Mn and γ 2.0 Gy and control groups.

Group		Hour6		Day3		Day14		Month2		Month6		Group	Month8	
Hemorrhage	^56^Mn(4×)	3^a)^	(+~++)^b)^	3	(++~+++)	2	(+)	3	(+~++)	4	(+~+++)	^56^Mn(1×)	6	(++~+++)
Emphysema		2	(++~+++)	2	(++)	2	(+~++)	3	(+~++)	4	(+~++)		6	(++~+++)
Inflammation		3	(+~+++)	3	(++)	2	(+~++)	3	(+~+++)	4	(++~+++)		6	(++~+++)
Atelectasis		0	(-)	0	(-)	0	(-)	0	(-)	2	(++~+++)		3	(++~+++)
Pneumonia		0	(-)	0	(-)	0	(-)	0	(-)	1	(+)		0	(-)
Granuloma		0	(-)	0	(-)	0	(-)	0	(-)	1	(+)		0	(-)
		(*n* = 3)		(*n* = 3)		(*n* = 3)		(*n* = 3)		(*n* = 4)			(*n* = 6)	
Hemorrhage	^56^Mn(2×)	3	(+~++)	3	(++~+++)	2	(+~++)	3	(++)	2	(+~++)	Mn	1	(+)
Emphysema		2	(+)	3	(+~++)	3	(+~+++)	3	(+~++)	2	(+~++)	stable	1	(+++)
Inflammation		3	(+~++)	3	(++~+++)	3	(+~+++)	3	(++)	2	(++~+++)		1	(+)
Atelectasis		0	(-)	1	(++)	0	(-)	0	(-)	1	(+)		1	(-)
Pneumonia		0	(-)	0	(-)	0	(-)	0	(-)	1	(+)		0	(-)
Granuloma		0	(-)	0	(-)	0	(-)	0	(-)	0	(-)		0	(-)
		(*n* = 3)		(*n* = 3)		(*n* = 3)		(*n* = 3)		(*n* = 3)			(*n* = 4)	
Hemorrhage	γ 2.0 Gy	3	(+~++)	0	(-)	1	(+)	1	(+)	1	(+)	γ 2.0Gy	1	(++)
Emphysema		2	(+~++)	2	(+)	0	(-)	1	(+)	0	(-)		2	(+)
Inflammation		3	(+)	1	(+)	1	(+)	1	(+)	1	(+)		2	(+~+++)
Atelectasis		0	(-)	0	(-)	0	(-)	0	(-)	0	(-)		1	(+)
Pneumonia		0	(-)	0	(-)	0	(-)	0	(-)	0	(-)		0	(-)
Granuloma		0	(-)	0	(-)	0	(-)	0	(-)	0	(-)		0	(-)
		(*n* = 3)		(*n* = 3)		(*n* = 3)		(*n* = 3)		(*n* = 4)			(*n* = 4)	
Hemorrhage	Control	0	(-)	0	(-)	0	(-)	0	(-)	0	(-)	Control	0	(-)
Emphysema		0	(-)	1	(+)	0	(-)	0	(-)	0	(-)		1	(+)
Inflammation		1	(+)	1	(+)	0	(-)	1	(+)	0	(-)		0	(-)
Atelectasis		0	(-)	0	(-)	0	(-)	0	(-)	0	(-)		1	(+)
Pneumonia		0	(-)	0	(-)	0	(-)	0	(-)	0	(-)		0	(-)
Granuloma		0	(-)	0	(-)	0	(-)	0	(-)	0	(-)		0	(-)
		(*n* = 3)		(*n* = 3)		(*n* = 3)		(*n* = 3)		(*n* = 4)			(*n* = 5)	

a) Number of rats with incidence and pathologic grades (in parenthesis). b) Pathologic grades were scored from − to +++.

**Table 3 biomedicines-08-00171-t003:** Absorbed dose rate and the accumulated dose surrounding the ^56^Mn particles.

Distance from Center (μm)	^56^Mn Initial Dose Rate (Gy/h)			Accumulated Dose of 6 h (Gy)			Accumulated Dose of ∞ (Gy)		
	Diameter (μm)			Diameter (μm)			Diameter (μm)		
	3	5	10	3	5	10	3	5	10
1.5	1.67			4.98			6.22		
2	7.21 × 10^−1^			2.15			2.68		
2.5	4.38 × 10^−1^	2.70		1.30	8.05		1.63	10.05	
3	2.91 × 10^−1^	1.52		8.67 × 10^−1^	4.52		1.08	5.64	
5	9.99 × 10^−2^	4.72 × 10^−1^	5.19	2.98 × 10^−1^	1.41	15.5	3.72 × 10^−1^	1.76	19.3
10	2.41 × 10^−2^	1.11 × 10^−1^	9.07 × 10^−1^	7.19 × 10^−2^	3.31 × 10^−1^	2.70	8.97 × 10^−2^	4.14 × 10^−1^	3.37
50	8.94 × 10^−4^	4.12 × 10^−3^	3.27 × 10^−2^	2.67 × 10^−3^	1.23 × 10^−2^	9.74 × 10^−2^	3.33 × 10^−3^	1.53 × 10^−2^	1.22 × 10^−1^
100	2.15 × 10^−4^	9.92 × 10^−4^	7.88 × 10^−3^	6.40 × 10^−4^	2.96 × 10^−3^	2.35 × 10^−2^	8.00 × 10^−4^	3.69 × 10^−3^	2.93 × 10^−2^
500	7.43 × 10^−6^	3.43 × 10^−5^	2.73 × 10^−4^	2.21 × 10^−5^	1.02 × 10^−4^	8.14 × 10^−4^	2.77 × 10^−5^	1.27 × 10^−4^	1.02 × 10^−3^
1000	1.60 × 10^−6^	7.38 × 10^−6^	5.87 × 10^−5^	4.76 × 10^−6^	2.20 × 10^−5^	1.75 × 10^−4^	5.95 × 10^−6^	2.74 × 10^−5^	2.18 × 10^−4^
2000	2.97 × 10^−7^	1.38 × 10^−6^	1.10 × 10^−5^	8.85 × 10^−7^	4.10 × 10^−6^	3.27 × 10^−5^	1.11 × 10^−6^	5.12 × 10^−6^	4.08 × 10^−5^
3000	1.03 × 10^−7^	4.77 × 10^−7^	3.80 × 10^−6^	3.07 × 10^−7^	1.42 × 10^−6^	1.13 × 10^−5^	3.84 × 10^−7^	1.78 × 10^−6^	1.42 × 10^−5^
5000	2.61 × 10^−8^	1.21 × 10^−7^	9.67 × 10^−7^	7.78 × 10^−8^	3.61 × 10^−7^	2.88 × 10^−6^	9.72 × 10^−8^	4.51 × 10^−7^	3.60 × 10^−6^
8000	5.17 × 10^−9^	2.39 × 10^−8^	1.90 × 10^−7^	1.54 × 10^−8^	7.12 × 10^−8^	5.67 × 10^−7^	1.92 × 10^−8^	8.90 × 10^−8^	7.08 × 10^−7^
10,000	1.47 × 10^−9^	6.73 × 10^−9^	5.38 × 10^−8^	4.39 × 10^−9^	2.01 × 10^−8^	1.60 × 10^−7^	5.48 × 10^−9^	2.51 × 10^−8^	2.00 × 10^−7^
12,000	2.28 × 10^−10^	1.03 × 10^−9^	8.21 × 10^−9^	6.79 × 10^−10^	3.08 × 10^−9^	2.45 × 10^−8^	8.48 × 10^−10^	3.84 × 10^−9^	3.05 × 10^−8^
13,000	4.57 × 10^−11^	2.11 × 10^−10^	1.65 × 10^−9^	1.36 × 10^−10^	6.29 × 10^−10^	4.91 × 10^−9^	1.70 × 10^−10^	7.85 × 10^−10^	6.13 × 10^−9^
14,000	2.17 × 10^−12^	1.00 × 10^−11^	7.39 × 10^−11^	6.47 × 10^−12^	2.98 × 10^−11^	2.20 × 10^−10^	8.07 × 10^−12^	3.72 × 10^−11^	2.75 × 10^−10^

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
