# Peer review of "Impact of Local High Doses of Radiation by Neutron Activated Mn Dioxide Powder in Rat Lungs: Protracted Pathologic Damage Initiated by Internal Exposure"

_biomedicines, 2020, doi:10.3390/biomedicines8060171_

Round 1
Reviewer 1 Report
The submitted manuscript entitled “Impact of locally ultra-high radiation by neutron activated Mn dioxide powder in rat lung: protracted pathological damage initiated by internal exposure” investigates the short and long-term effects of internal exposure of 56MnO2 on the lung. The authors propose increased elastin as an earlier indicator of radiation induced lung damage. While the data presented is novel and interesting, there are a number of concerns.
The manuscript would benefit overall from extensive editing. There are numerous errors throughout (some examples; line 51 “Due to its short physical half-life ~of 2.58 h…”, line 228 “Apoptosis was ~clearly evident…”, line 321 “Tissue ~cells…”, line 346 2.75 should be 2.74.). All instances of 56Mn dose need to be corrected (2.74x108). Adjustments need to be made to Table 1 and 2, which may have shifted during submission, particularly Table 1. Appropriate headings are required for all columns. There are consistency issues throughout the manuscript, ex. beta vs β, 8 months vs day 240. Also, sentences are presented as separate statements, with no clear connection. For example, in the Discussion, line 242-250, the paragraph begins stating high dose induced pulmonary fibrosis and pneumonitis. The following sentence “Here we show late stage data for internal exposure”, however, then discuss early event induction of elastin. The sentence “Here we show late stage data for internal exposure” does not appear to belong in this paragraph. The preceding paragraph discusses Type II pneumocytes, but the concluding sentence defines basal cells. Though this sentence is relevant to the manuscript, it does not belong in this paragraph. There are other instances where statements are presented without a clear connection to the surrounding text.
Throughout the manuscript, the exposure dose is described as low average dose ultra-high local dose and dose rate. The authors use these doses interchangeably throughout the manuscript when describing dose effects on outcomes measured. As these doses are very different (ultra-high to low), is this appropriate? If the localized dose is ultra-high, is the average dose to the whole tissue relevant? The authors even state in the Discussion “A pathological progression potentially leading to mortality is not likely dependent on average accumulated dose level, but rather on activation level of the 56Mn particle.” Yet, the authors frequently state the dose of 0.10 Gy is ‘responsible’ for the outcomes measured. The manuscript will benefit if the authors mainly focus on a single dose or further highlight the importance of the localized dose vs average dose.
In the second paragraph of the Introduction, line 60-62 “The impact of internal exposure to organ tissue at low tissue average doses was equivalent to high dose effects from external exposure. The previous findings were shown in the data of lung tissue for hemorrhage, emphysema and inflammation of 3 days and 60 days after 100 mGy internal exposure”, there is no reference for these statements.
In the Results, the 56Mn doses are presented as 1x2.74x108, 2x2.74x108, 4x2.74x108, it is unclear if for the higher dose groups rats were exposed 2 or 4 times or a single exposure of double or quadruple the original dose. If the latter, 56Mnx4 and 56Mnx2 should instead be presented as 1.10×109 Bq and 5.48×108 Bq respectively.
It is evident by Table 3 that the majority of 56Mn exposure was in the stomach and intestines, yet there is no mention of this in the manuscript.
It is not specified which lung lobe was used for images and analysis, or if all lobes were analyzed and combined. If so, was it necessary to use whole lung, why not use a portion for wet:dry measurements for edema, which would further confirm lung injury.
In Figure 4 it appears Mn stable has more collagen staining than the others.
In the Discussion it is specified the majority of cells were neutrophils and macrophages but there is no mention of how these cells were differentiated.
In line 227 it is stated “…stem cell apoptotic damage with normal body weight…” What is the relevance of weight? Are number of apoptotic cells normalized to body weight? There is no mention of weight measurements in the manuscript.
The authors state the accumulation of Fe is likely due to a blood clot, however, could this not be confirmed with H&E staining? Are there red blood cells where increased Fe was observed?
Line 299 “Although previous research has shown pathological changes in the lungs from an external dose of more than 8 Gy…” Reference?
Line 312 “Apoptosis occurring in the bronchiole is likely an indicator of stem cell damage” could the authors describe why this is likely.
Line 314 “Apoptotic stem cell damage for internal exposure continued through 8 months evidence? and it suggests that epithelial repair mechanisms are compromised by damaged DNA signatures”
Is there any evidence to suggests if inflammatory cells uptake (phagocytize) 56Mn in the lung? Has 56Mn been measured in the blood? Table 3 shows 56Mn exposure in various tissues, possibly due to travel 56Mn traveling through blood.
The manuscript would benefit from discussion of the limitations of the study, such as the relatively small group numbers (n=3).
Briefly define Mn stable.
There is no mention of dissection or tissue collection in the Methods section.
The manuscript dose not appear to have a conclusion.
Reviewer 2 Report
This manuscript describes short- and long-term pathological changes in the lungs of male rats exposed to 56MnO2 particles and compares these exposures with the effects of a single dose of external gamma-rays. While the research is relevant, many clarifications need to be made in the text, as outlined below.
- Title: Please change “locally ultra-high radiation” into “locally delivered radiation” or “local high doses of radiation”.
- Abstract:
- Please indicate the strain and sex of the rats.
- Please mention that internal exposures were compared with external gamma-rays.
- Results:
- To orient the reader, in the first paragraph of the results section, please describe what 2x and 4x2.74x108 Bq stand for. Do they mean 1 administration at 2 and 4 times the activity, or are they 2 and 4 separate administrations?
- Line 101: elastic fibers are “broken at intervals”. What do the authors mean? Please indicate in Figure 4 how we can identify these broken intervals.
- Lines 119-120: What does “fatal hypofunction” mean? The manuscript does not describe any lung function tests.
- It looks like apoptosis was assessed only at 60 days after inhalation. Why was only this late time point included, when apoptosis is thought to occur within the first few days-weeks after irradiation?
- Line 164: Should “Figure 5B” be corrected into “Figure 6B”?
- Paragraph 2.4 seems to repeat some of the information in paragraph 2.3. For clarity, it may be good to combine these two paragraphs and remove the duplicate information.
- In Figure 1, please use arrows in the pictures to indicate the pathological findings that are described in the figure legend.
- In Figure 2, the X-axes: What does “e1-240d” mean?
- In Figure 3, please use arrows in the pictures to indicate the pathological findings that are described in the figure legend.
- The legend of figure 3 states that severe inflammatory cell infiltration is seen. This is difficult to see in the pictures. Some pictures at a higher magnification may help.
- The legend of Figure 4 states that the 56Mn rat was exposed to 0.10 Gy. This dosage is not specified anywhere else in the manuscript. It seems to come from Table 3 (from a prior publication). It may be better to stay consistent with the rest of the manuscript and call this 56Mnx1 (correct?).
- Figure 5 does not seem necessary. The range of the particle sizes can be described in the methods section.
- Figure 6:
- The text in Figure 6B and Figure 6C is too small and cannot be read.
- Is the absorbed dose rate shown in Figure 6B immediately upon inhalation of the 56Mn? Please clarify in the figure legend.
- Concentrations are indicated as ug/g. Should this be microg/g?
- Table 1:
- Should the words on the left (Hemorrhage, Emphysema etc.) all be lined up to the left of the table?
- Please explain the meaning of the numbers (6, 6, 6, 3, 1, 1, 1, 1, etc.) in the third column of the table.
- Table 2: It is unclear what the titles “Diameter” and “Distance” refer to. Please restructure the table to make it easier to read.
- Table 3 seems to be a copy of an already published table in reference 5. For the current study, only the lung absorbed doses seem relevant. Please remove the table and provide the lung dose information somewhere in the text.
- Discussion:
- Line 229 describes apoptosis at 8 months after 56Mn inhalation. However, the results seem to show that apoptosis was assessed only at 60 days (2 months).
- Lines 231-241: The purpose of this paragraph in relation to the findings in the current manuscript is not clear.
- Line 243: This study exposed animals to 5 Gy whole body followed by 10 Gy to the lungs (not 15 Gy of fractionated exposure). Please correct in the text.
- Lines 252-253 explain that 0.11 Gy of 56Mn (to the lungs?) is roughly equivalent to an external gamma-ray dose of 2 Gy. This is important information to justify why the authors have selected an external dose of 2 Gy. Please provide this information earlier in the manuscript (for instance early on the results section). Also, please explain where the 0.11 Gy 56Mn comes from. And also explain who this conversion was calculated (based on RBE of local beta-rays compared to external gamma-rays?).
- Lines 255-256: “the effect of 56Mn internal exposure is similar in effect to that of a 20 times higher dose of gamma external exposure”. However, lines 262-263 describe that the internal exposure has a unique pathology. These two statements seem to contradict each other.
- Lines 291-292: “56Mnx1 group, the same accumulated absorbed dose in lung tissue results in severe pathological findings in 56Mnx4 group”. It is not clear what the authors are trying to say in this sentence. Also, does the same accumulated absorbed dose come from Table 3? Please clarify in the text.
- Lines 298-310: The only information in this paragraph that seems relevant is the information in reference 35 (lines 303-304).
- Lines 311-317: The information in this paragraph does not seem relevant to the study. Early apoptosis (within weeks after radiation) was not examined. Stem cells were not examined.
- Lines 321-331: The information in this paragraph does not seem relevant to the study.
- Methods:
- Line 347: Is reference 46 correct? It seems to refer to a nuclear rocket engine reactor (not a rat exposure box).
- Paragraph 4.2: Please explain what 2x and 4x 56Mn stand for. If this is one inhalation of 56Mn at a higher activity, then the production of this 56Mn needs to be described in paragraph 4.1.
- Lines 357-361: Please move these lines to the beginning of the paragraph.
- Section 5. Conclusions seems missing.
- Throughout the manuscript, 56Mn activity is indicated as ...x108 Bq or ...x109 Bq. Please make sure to put the 8 and 9 into superscript.
- The manuscript has many mistakes in the English language and is therefore sometimes difficult to understand. Please ask a native English speaker to help edit the text.
Author Response
Please see the attchment.

Round 2
Reviewer 1 Report
The author's adequately addressed the reviewers concerns and I can now recommend this manuscript be accepted for publication. There are a few minor spelling issues that can be fixed during the author's final edit of the proofs (Figure 1D, months spelled incorrectly).
Reviewer 2 Report
The authors have made many changes I have asked for. I think it is important to still do the following: 1. In the legend of figure 4A, describe that the arrows indicate broken elastin fibers. 2. The manuscript still has several errors in the English language and needs language editing.This manuscript is a resubmission of an earlier submission. The following is a list of the peer review reports and author responses from that submission.
Round 1
Reviewer 1 Report
Shichijo et al. examined the Mn localization in lung after internal exposure of rats to 56MnO2 powder for local dose effects by β rays from 56Mn. Pathological evidences by local dose demonstrated in this paper are valuable for refinement of dose evaluation. However, the information about importance of 56Mn in dose evaluation after nuclear accidents was insufficient and quantitative data on Mn localization were missing. The introduction gives the impression of confusing the present paper with previous works of the authors. The research area of this paper is restricted and unsounds for the readers of International Journal of Molecular Sciences. The authors might consider submitting this paper to a journal specializing in radiation science topics, such as Health Physics or Radiation Research.
Reviewer 2 Report
Comment on the manuscript ijms-756458 “Impact of locally ultra-high radiation by 56Mn dioxide powder in rat lung: protracted pathological damage initiated by internal exposure”
by K. Shichijo et al.
The manuscript contains important results which are useful for analysis of consequences of nuclear disasters of several kinds, treatment of the sufferers and can be useful for optimization of some scenarios of radiation therapy and diagnostics. However, publication of the manuscript will be possible only after a modest revision the necessity of which results from the following:
- The authors neither discuss the possibility of enhancement of the biological effect of electrons, emitted by 56Mn, by particles of MnO2 nor state directly that they consider such enhancement unimportant. The importance of this enhancement in the experiments described in the manuscript is unclear. It would be highly desirable to use gamma rays in combination with the use of nonactivated 55MnO2 Currently, the possibility of enhancement of biological effect of ionizing radiation by nonradioactive materials, in particular, by nanoparticles is discussed very widely (the fact that the authors used microparticles is not principally important, because the use of nanoparticles in experiments and theoretical studies of other researches is determined by medical problems, while more or less similar enhancement can also be caused by microparticles). Studies of this enhancement are mainly devoted to radiation therapy, but the possibility of its manifestation in other situations, for example, at the use of Thorotrast was also discussed (see e.g. M.L. Shmatov, Phys. Part. Nucl. Lett. 13, 514 (2016)). It should be emphasized that this enhancement is possible even in the situations when an increase in physical dose is negligible (see e.g. K. Kim et al., Phys. Med. Biol. 57, 8309 (2012); T. Wolfe et al., Nanomedicine 11, 1277 (2015); J. Schuemann et al., Int. J. Rad. Oncol. Biol. Phys. 94, 189 (2016); L.E. Taggart et al., Cancer Nanotechnol. 5, 5 (2014); M.L. Shmatov, Phys. Part. Nucl. Lett. 14, 533 (2017); M. Fuss et al., Phys. Med. Biol. https://doi.org/10.1088/1361-6560/ab7504 (2020) (available online)). Therefore, its manifestation in the experiments of the authors is potentially possible. If the authors cannot perform experiments with gamma rays and nonactivated 55MnO2 particles in the observable future, it is necessary to mention directly the fact that the problem of such enhancement was not studied and to cite several proper papers.
- The text should be edited:
- The title of the manuscript and chemical formula 56MnO2 on page 10 allow us to assume that after activation, MnO2 contained almost pure 56 If this assumption is correct, the interesting fact of the use of almost pure 56Mn should be mentioned, while if it is mistaken, the real initial isotope composition of Mn should be presented and the title of the manuscript should be changed.
- At the end of 1st sentence of Introduction, the author write about “the initial radiation directly received from the bombs”, while the beginning of the sentence corresponds to “the initial radiation directly received from the bombs or other sources” or something similar. In the next sentence, it is necessary to write “or other nuclear disasters” or something similar after the word “detonations”.
- Notations “56Mn x 4”, etc. are used for the first time on page 4 and explained only on page 17. The explanation should be presented before or just after the first use of the notations.
- The use of the word “Surprisingly” in the first sentence on page 4 is unclear, because delayed effects of ionizing radiation are known (see also the first sentence of Section 3, page 13). If my statement is not correct, the authors should explain the use of this word in their Reply to Comment. The next sentence should probably be a part of the previous one.
- Heading of subsection 2.3 and two first sentences of this subsection (page 9) should be edited. In particular, the facts that the word “diameter” corresponds to oxide particle and distance is measured from its center should be mentioned. Table 2 does not show spectrum (see also item 2.9).
- The terms “56Mn particles” and “Mn particles” (pages 9 and 10) are unclear.
- The sentence “We have already reported the frequency distribution of MnO2 powder particle sizes used in this experiment [4]” (page 10) is formally correct, but for convenience of readers, it is desirable to present the distribution again. In any case, the use of the word “frequency” is unclear, I think that it is necessary to write “size distribution” or something similar.
- The words “is in the process of ejection” (page 11) are unclear.
- 5C does not contain information about energy spectrum; “Distance from Center” is measured in micrometers, not in millimeters.
- When mentioning dose rates around particles, it is necessary to mention whether deceleration of electrons in particles was taken into account (if this effect is negligible, this should be written).
- The sentence “A 56Mn particle that has a valence of 4 chemical species as MnO2 changed to a valence of 2 after deposition in lung tissue (Figure 6)” is unclear.
- The sentence on pages 15-16 is unclear.
- The words “including Mn stable” (page 17) are unclear.
Round 2
Reviewer 1 Report
My judgment is the same as the first one.
I recommend the authors to submit this paper to a journal of radiological sciences.
